# Dispersion of particulate matter (PM$_{2.5}$) from wood combustion for residential heating: Optimisation of mitigation actions based on large-eddy simulations

Tobias Wolf[1], Lasse H. Pettersson[1], Igor Esau[1]

[1]Nansen Environmental and Remote Sensing Center, Thormøhlens gate 47, 5006, Bergen, Norway

*Correspondence to*: Igor Esau (igore@nersc.no)

**Abstract.** Many cities in the world experience significant air pollution from residential wood combustion. Such an advection-diffusion problem as applied to geographically distributed small-scale pollution sources presently does not have a satisfactory theoretical or modelling solution. For example, statistical models do not allow for pollution accumulation in local stagnation

zones – a type of phenomena that is commonly observed over complex terrain. This study applies a Parallelized Atmospheric Large-eddy simulation Model (PALM) to investigate dynamical phenomena that control variability and pathways of the atmospheric pollution emitted by wood-burning household stoves. The model PALM runs at spatial resolution of 10 m in an urban-sized modelling domain of 29 km by 35 km with a real spatial distribution of the pollution source and with realistic surface boundary conditions that characterize a medium-sized urban area fragmented by water bodies and hills. Such complex

geography is expected to favour local air quality hazards, which makes this study of general interest. The case study here is based on winter conditions in Bergen, Norway. We investigate the turbulent diffusion of a passive scalar associated with small sized particles (PM2.5) emitted by household stoves. The study considers air pollution effects that could be observed under different policy scenarios of stove replacement; modern wood stoves emit significantly less PM2.5 than the older ones, but replacement of stoves is costly and challenging process.

We found significant accumulation of near-surface pollution in the local stagnation zones. The simulated concentrations were larger than the concentrations obtained only due to the local PM2.5 emission, thus, indicating dominant trans-boundary contribution of pollutants for other districts. We demonstrate how the source of critical pollution can be attributed through

model disaggregation of emission from specific districts. The study reveals a decisive role of local air circulations over complex terrain that makes high-resolution modelling indispensable for adequate management of the urban air quality.

This modelling study has important policy-related implications. Uneven spatial distribution of the pollutants suggests prioritizing certain limited urban districts in policy scenarios. We show that focused efforts towards stove replacement in specific areas may have dominant positive effect on the air quality in the whole municipality. The case study identifies urban districts where limited incentives would result in the strongest reduction of the population's exposure to PM2.5.

## 1 Introduction

Residential wood combustion in households is a significant global air polluter. Even in developed countries, e.g., in Scandinavia, wood burning for residential heating is a dominant source of air pollution in cold winter days (Savolahti et al., 2019; Kukkonen et al., 2020). A low temperature wood-burning process emits a considerable amount of particulate matter of less than 2.5 micro-meter (µm) in size. Such particles are collectively abbreviated as PM2.5. In fact, residential combustion is the largest PM2.5 emitter in the Nordic countries, including Norway (Im et al., 2019). Studies reveal that PM2.5 concentrations

attributed to residential wood combustion in Oslo may reach 60% of the total annual average concentrations of this pollutant (Kukkonen et al., 2020). The attributed fraction of PM2.5 can be even larger in smaller cities (Hedberg et al., 2006). In the short run, air pollution during cold winter days can result in reduced visibility and pervasive smoky smell in the air. The highest levels of pollution are found in the most densely populated central urban districts where wood-burning is used in apartments of multi-story houses. The PM2.5 concentrations exceeding 40 $\mu gm^{-3}$ can lead to an increased number of health issues in

exposed populations. About 2000 annual premature mortality cases have been attributed to PM2.5 pollution in Norway alone (Im et al., 2019). Overall, non-industrial wood combustion in Europe contributes to about 10% of the total health cost of air pollution (Brandt et al., 2013). These numbers, as well as persistent public pressure and regulations from environmental protection agencies, are pushing many countries in the world, including Nordic municipalities, to introduce policy measures fostering reduction of residential emissions.


To be effective and efficient, policy measures must be supported by science. Atmospheric pollution caused by small-scale spatially distributed emitters is a non-trivial advection-diffusion problem that does not have a general solution. It is typically dealt with using numerical modelling. However, the most common Gaussian statistical models are unable to represent pollution aggravation in local stagnation zones. This problem is related to the fact that pollution dispersion models use only a diffusion

operator to calculate the concentrations along the pollution transport pathways. The worst air quality is typically observed in atmospheric conditions of weak turbulent mixing where horizontal advection dominates over lateral diffusion. In such conditions, pockets of highly polluted air can be transported over long distances, eventually exacerbating the air quality problems in remote stagnation zones downstream. It is important to emphasize that local emission in the stagnation zones may

be less significant, so that, those zones are not identified by statistical models as being vulnerable to air quality hazards.

Turbulence-resolving atmospheric models have been found to be a proper and general tool to study passive scalar dispersion over complex terrain (Gousseau et al., 2015; Zhong et al., 2016). Such models simulate turbulence diffusion and pollution transport on the basis of air flow dynamics. Up to now, however, realistic studies with the turbulence-resolving models were limited as they require significant computing resources. This study is one of the first that assesses air quality and supports policy development for the whole urban municipality with hydrodynamical simulations that resolve all significant buildings.


Air quality is controlled by near-surface turbulent mixing and local winds. We treat small smoke particles (PM2.5) as passive scalars emitted to the atmosphere at actual locations of household chimneys. Each chimney creates high PM2.5 concentrations only in its immediate vicinity. A statistical model adds up these local emissions making the total concentrations roughly proportional to the chimney density and the local emission rates (Zhong et al., 2016). With our turbulence-resolving model,

we aim to understand contributions from the dynamical component of turbulent diffusion and transport in spatial variability of the observable PM2.5 concentrations. In realistic urban domains such as, e.g., in Bergen, Norway, concentrations are patchy and follow intricate pattern of local wind convergence and stagnation zones (Wolf et al., 2020). Such microscale climate information must be considered in air quality impact studies (Bai, 2018; Chandler, 1976). A challenge is however related to modelling of local winds and turbulence patterns. A turbulence-resolving model must be run with a realistic inventory of

household emission sources and with correct topography in a sufficiently large domain.

One might argue that a dense network of low-cost meteorological and air quality sensors can do the job when it is combined with statistical models (Schneider et al., 2017) or machine learning (Venter et al., 2020). That is likely to be true. Policy makers, however, would benefit from the model-based approach as it provides quantitative evaluations of the impact of proposed measures, even in the case when those measures are changing configurations and intensities of emission sources.

Ultimately, as we will demonstrate in this study, the models can help to identify urban areas vulnerable to air pollution, and to justify more optimal policies complying with targeted air quality levels. Locked within a silo (i.e., vertically integrated) approach to environmental management, Nordic municipalities are reluctant to explore advanced modelling approaches (Leiren and Jacobsen, 2018). In such circumstances, a realistic case study is beneficial to demonstrate the added value of

turbulence-resolving modelling for air quality management. We chose a domain of the Bergen municipality for such a

prototype demonstration study.

Actionable policy scenarios must reflect the complex physio-geographical context of urban areas. Hence, the models of atmospheric chemistry and physics must include effects of local topography and spatial distribution of population and pollution sources. A specific interest for research is related to long-range pollution transport where the strongest air quality effects are observed further away from the source of emission. The long-range transport in a local sense is a transport over a few hundred

and thousands of meters. Such effects are typical in valleys or in complex coastal areas with significant local circulations (Fernando et al., 2010). Numerical weather prediction models with their typical geographical resolution of 1 km or coarser cannot represent this complex environment (Baklanov et al., 2017a; Køltzow et al., 2019). In addition, large buildings, roads systems and other infrastructure objects modify winds and turbulent dispersion shading some places or aggravating pollution in others (Gousseau et al., 2015). A growing body of literature suggests that turbulence-resolving or at least turbulence-

permitting large-eddy simulation (LES) models are needed to deal with the urban pollution issues (Baklanov et al., 2017b; Brötz et al., 2014; Grimmond et al., 2020; Kurppa et al., 2018; Stoll et al., 2020). Fine-scale LES is a computational fluid dynamics technique that becomes affordable with growing performance of massive-parallel computers. The LES models resolve the most energetic and dispersive turbulent eddies in the stratified lower atmosphere over realistically complex surfaces.

The LES models have already been used to simulate turbulent flows and atmospheric pollution in several smaller urban areas (e.g., Castillo et al., 2009; Cécé et al., 2016; Gronemeier et al., 2017; Keck et al., 2014; Letzel et al., 2008; Park et al., 2015b; Resler et al., 2017a).Our study further extends LES modeling of the air quality transport and diffusion problem to a whole municipality. We run the model with real surface boundary conditions, atmospheric meteorological conditions, and actual distribution and effect of emission sources. In this configuration, the modeling results could be used in policy scenario

evaluation and decision-making processes. We use the Parallelize Atmospheric Large-eddy simulation Model (PALM) described by Maronga et al. ( 2015, 2019b) with minor own modifications as described by Wolf et al.( 2017a, 2020). Furthermore, we investigate a set of plausible mitigation policy scenarios, which were proposed to reduce emissions from the residential wood combustion sources – the household stoves. The central policy measure is to push for replacement of older

stoves to less polluting new stoves. This push is given through soft economic incentives – a limited cash refund of 5000 NOK

per stove – and unconditional hard policy stipulating a ban on the use of the older stoves by 2021. The policy was designed

primarily fire safety considerations in mind, as well as improving the air quality. However, its effect on air quality and people's

exposure to high PM2.5 concentrations have not been quantitatively evaluated before this study.

The manuscript has the following structure. The next section describes the context, geography, and datasets of this study. The

third section describes the PALM model, setup of the simulations, and our methodology. The fourth section presents the

obtained results. The fifth section provides a broader discussion with generalizations of the methodology, data usage and

potentials for policy implications. The final section summarizes the conclusions.

## 2 The Bergen case study: Local context and datasets

Bergen is the second largest city in Norway. Its population is more than 275,000 people. It occupies over 465 km$^2$ divided into

8 districts. Lower parts of hills and coastal valleys within the Bergen municipality are built up. Central urban districts and

115 several neighbourhoods around large shopping malls are densely populated and include high-rise residential and administrative

buildings. The central urban districts have a population of more than 75,000. They occupy a narrow Bergen valley, which

opens on both sides towards large sea inlets (fjords). Topographic sheltering (Jonassen et al., 2013) by surrounding mountains

up to 643 meters height, together with maritime boreal climate at 60.4°N, favour persistent winter-time surface temperature

inversions capping cold air pools in the lowest parts of relief (Lareau et al., 2013). The temperature inversions, typically about

120 250 meters height, can last over several days. They are robustly associated with clear sky, calm and cold weather conditions

(Wolf et al., 2014).

As in many Nordic cities, westerly winds bring clean air into Bergen from the Atlantic Ocean. The main local permanent

PM2.5 sources are ships in the harbor and road traffic (Wolf et al., 2020). In winter months, however, households actively use

wood-burning stoves as a secondary heating source. The household stoves considerably contribute to the PM2.5 pollution in

125 Bergen in November through April. Accurate inventory of the residential PM2.5 emission is nevertheless problematic as we

will discuss later in our study. Ample use of wood-burning frequently leads to high ambient concentration of PM2.5 that in

several urban districts exceeds a threshold of 40 $\mu gm^{-3}$ set by the environmental protection agency (Høiskar et al., 2017; Wolf

et al., 2020). A large fraction of PM2.5 emission comes out from household wood-burning stoves installed before 1998, hereafter referred to as older stoves. These older stoves have much higher emissions of particulate matter per unit volume of wood; and they are less combustion and energy efficient than more modern "clean-burning" stoves, which we will refer to as new stoves. There is a strong anti-correlation between air quality and air temperature in Bergen (Wolf and Esau, 2014). Low air temperatures are associated with calm weather periods when residential wood-burning is enhanced and the emitted particles are trapped in the urban canopy due to weak vertical turbulent diffusion in the strongly, often inversely stratified lower atmosphere (Wolf et al., 2014). Although turbulent diffusion is weak, local near-surface micro-circulations (local winds) develop within this highly heterogeneous urban domain (Wolf-Grosse et al., 2017a). The local winds aggravate air pollution in convergence and stagnation zones. Thus, a general advection-diffusion problem translates in the Bergen context into the concrete modeling task of identification of urban districts with elevated PM2.5 concentrations given a realistic distribution of household stoves, emission scenarios and weather conditions.

This study addresses the following questions. What is the spatial distribution of PM2.5 concentrations in the city during critical air pollution episodes? What is the impact of emissions in each district on the (overall) air quality in the most polluted parts of the city? What would be the effect of replacing the existing older stoves with only clean burning (new) stoves in the entire city or in some of its districts? These research questions were formulated by scientists. Policy makers and stakeholders might enrich the study providing for different opportunities and stove replacement strategies. The study benefits from communication with the Bergen municipality and other interested parties. Many Nordic cities are suffering from similar challenges, and this is today an active field of research (Simpson et al., 2018).

## 3 The model and method

### 3.1 The large-eddy simulation model PALM

We simulate the lower atmosphere (up to 2.2 km) over the whole Bergen municipality with the LES model PALM. PALM version 5.0 (revision 3063) is an atmospheric large-eddy simulation model developed by the PALM group at the Leibnitz University of Hannover, Germany (Maronga et al., 2015). The model solves primitive hydro- and thermo-dynamic equations for incompressible, Boussinesq fluids. PALM explicitly resolves a part of relevant three-dimensional atmospheric dynamics

as well as turbulence-flow interactions with complex surface geometry. These features give some advantages to the LES-based approach as compared to a more traditional meso-meteorological modelling in the urban areas. Our reader shall, however, observe that we do not claim accurate simulations of details of such interactions. We only expect that PALM is robust for the

dominant scales of the relief (hundreds of meters in Bergen), which are well resolved by the model grid.

Our own user-code in PALM aims to improve treatment of the complex surface and environmental conditions. This user-code includes a routine to relax the mean vertical temperature profile in the computational domain towards a given input profile. In this way, the model preserves the temperature inversion that traps the emitted pollutants within urban canopy. The user-code allows for mixed-type boundary conditions for the surface temperature; the von Neuman constant flux boundary conditions

are set over land, the Dirichlet constant temperature conditions – over water. Our user-code also re-calculates temperature and concentrations of pollutants in terrain-following coordinate system from the model output provided on the rectangular grid. For this, a linear extrapolation is conducted, if the specified height above the ground is not a multiple of the grid height at the given grid-point or simply the variable at the correct grid-height is given.

Pollutants are treated as passive scalars. Chemical reactions are not simulated. Household chimneys are not resolved in the model. To compensate for variable heights of the houses and their chimneys, the user-code allows for emission at any height above the surface. The emission height is then given as input to the model as an array of numbers for each horizontal grid point. The passive scalar is assumed being instantaneously mixed within the entire grid cell of emission. We apply a methodology of artificial pollutants prescribing emission of an independent pollution specimen at each district of choice. Thus,

pollution from each district remains independent and traceable over the whole domain. The lateral boundary conditions do not allow for recirculation and inflow of passive scalars into the model domain through any boundary. The three first methodologies are described in Wolf-Grosse et al. (2017) and Wolf et al. (2020) and the latter has been developed for this study.

### 3.2 The model setup for the Bergen case study

The model setup for the Bergen case study is discussed in details in Wolf et al. ( 2020), where it has been applied to study air pollution from a mixture of diverse (localized and spatially distributed) emission sources. Only a short summary and important

differences are presented below. The computational domain includes Bergen and parts of the surrounding municipalities (see Fig. 1). The total domain size is 28800 x 34560 m$^2$ in the zonal and meridional directions, respectively. The simulations run with horizontally periodic lateral boundary conditions for wind, pressure, and temperature. Therefore, this domain includes buffer zones with a width of 1000 m each, which are needed for linear interpolation between the opposing periodic boundaries. The horizontal grid resolution is 10 m. It gives a mesh of almost one million grid cells at each model level – the largest achieved urban simulations so far. The vertical grid resolution is 10 m up to 660 m height. Above that the vertical grid space increases by 1% for each additional grid level. The total domain height is 160 levels or 2239 m. Thus, the model domain top is found well above the highest hill (640 m) in the area of simulations. The surface in the model is approximated with cubes of 10 m x 10 m x 10 m . The topographic data for the approximation are taken from a laser-scan digital elevation model (DEM) of 1 m resolution provided by the Norwegian mapping authority (Statens Kartverk, 2018). At this resolution, DEM includes all buildings and trees in the city. DEM was delivered to us in the GeoTIFF format. We use the ArcGIS © software to process the data and to create a complete topography dataset of the required 10 m horizontal resolution. We fill in small gaps in the original DEM using the standard linear interpolation.

Water bodies have a distinct impact on boundary layer circulations, also in urban areas (Ronda et al., 2017; Wolf-Grosse et al., 2017a). This impact is accounted for in simulations by prescribing a constant negative surface heat flux of -20 W m$^{-2}$ over land. Constant surface temperature of 275.65 K (equal to 2.5$^o$ C) is applied over sea water; temperature of 273.15 K (equal to 0$^o$ C) is applied over freshwater bodies (lakes). This setup reflects conditions of winter temperature inversions with radiative surface cooling over land and warmer sea surface temperatures.

**3.3 Weather scenarios**

Damaging levels of air pollution are almost exclusively observed during persistent calm and cold weather episodes. We have already identified typical meteorological conditions that correspond to the pollution episodes (Wolf et al., 2014). Analysis of such conditions produced two influential meteorological scenarios, i.e., two sets of vertical profiles; one for temperature (from the meteorological temperature profiler in Bergen) and another for wind (from the ERA-Interim reanalysis) that correspond to the observed high concentrations of PM2.5. The first scenario is for the most typical winter conditions with high air pollution. This scenario is determined by the south-easterly geostrophic wind, which is used to force the model simulations, and by the

fjord surface water temperature of 2.5° C, which is used as the surface boundary condition in the model. In this scenario, the local winds transport polluted air from land to the city fjord through the densely populated central urban valley. The second meteorological scenario has: the easterly geostrophic wind of slightly higher speed; the fjord surface water temperature of 0°

C, which is still above the freezing point for salt water. Both scenarios are described in details in ( 2020) under the abbreviations ws01_wd01_ft01 and ws03_wd02_ft03, respectively. Model forcing and surface temperatures are fixed in both scenarios for the complete duration of the model runs. The runs are initialised with 12-hours long precursor runs. At this point, most meteorological parameters do not show any significant drift. Simulations with different configurations of emission sources are then conducted following the precursor runs for another 6 hours.

Positive turbulent sensible heat fluxes over water and negative fluxes over land are not in balance and do not cancel each other within the model domain. This imbalance, if not treated, would erode the stable stratification of the atmosphere in the upper layers of the model domain. We resolve the problem relaxing the mean temperature profile to the initially prescribed profile using nudging. Nudging corrects the temperature at every grid point and time step by a weighted difference between the actual domain-averaged temperature and that at the model initialisation. The weight is given by the relaxation time scale set to 43200

s at elevations below 400 m, which effectively gives no nudging in the lower layers of the model domain. The relaxation time scale linearly decreases till 1800 s at 600 m and higher elevations. Nudging is not applied at the first grid point above the surface.

We run PALM with real spatial configuration of small-scale emission sources in the case study domain that is provided to us by the Bergen Fire Brigade (Bergen brannvesen). Overall, there were 80 506 household stoves registered in Bergen in June

2018. The new clean burning stoves constitute 45.8% or 36 864 stoves. Most of them were installed in the recent years. Since 1995, only new stoves are allowed. We have aggregated and averaged the number of stoves within 3 x 3 grid cells plots (90 m$^2$) to emulate emission dispersion immediately after exhaust from chimneys. Figure 1 (bottom) shows the geographical distribution of the resulting emission source density. Concentrations are set to zero at starting both the scenario and sensitivity runs. In this study, we apply the same fixed emission height of 15 m (the 2$^{nd}$ model level) above the ground everywhere in the

model domain as it corresponds to the most typical building height in Bergen. Plausible effects of different emission heights are considered in a sensitivity study that is included in the supplementary material.

The Bergen Fire Brigade has also provided us with stipulated emission rates per stove. The new clean wood-burning stoves have the relative emission rate of 7 g kg$^{-1}$. This number shall be understood as 7 g of PM2.5 per kg of burned wood. The older stoves have the relative emission rate of 30 g kg$^{-1}$. With the given mix of the new and older stoves, the average relative emission rate is 19.42 g kg$^{-1}$. To obtain the actual emission rates, one must know consumption of wood per unit time per stove and frequency of using the stoves, i.e., parameters that characterize burning habits in the city. Felius et al. ( 2019) published an updated study of the wood-burning habits in Norway. It reveals that about 80% of households use stoves only during the afternoon and evening hours, while 90% of households still use wood-burning as a significant (primary and secondary) heating source. The typical average usage time is 4 to 10 hours. Shorter usage time (2.5 to 3.5 hours) is reported by Wyss et al. ( 2016) in a study of relations between indoor PM2.5 concentrations and wood-burning habits in Norwegian households. Emission rates also depend on the wood quality, wetness, and tree species in the wood supply (Hellén et al., 2008). These numbers are not available so that we estimated them using a small sampling study that is based on our social networks. We assume that each stove is used to burn approximately 11 kg wood per day, and that the wood consumption is 1.25 kg per hour (Solli et al., 2009), we arrived to 8.8 hours of wood burning, primarily between 15:00 and 24:00 local time; 90% of all aerosol exhaust comes out as PM2.5. Taken together, these assumptions and estimations result in the average PM2.5 emission rate of 0.7647 g s$^{-1}$ per stove. This number is about 10 times smaller than it has been assumed earlier by the Bergen Fire Brigade, but simulations with this emission rate shows good agreement between computed PM2.5 concentrations and the concentrations measured in the city. During the second week of February 2021, a persistent cold weather situation similar to our first scenario was observed in Bergen. The maximum simulated (observed on 11.02.2021) PM2.5 concentrations were 76.7 µg m$^{-3}$ (81.2 µg m$^{-3}$) at Danmarksplass, 53.4 µg m$^{-3}$ (59.2 µg m$^{-3}$) at Klosterhaugen in the city center, and 26.1 µg m$^{-3}$ (18.6 µg m$^{-3}$) at Rådal. This agreement demonstrates reasonably good capture of the spatial variability and accumulation of PM2.5 in the scenario simulations despite the accepted simplifications, assumptions, and uncertainties.

## 4 Results

Air pollution in urban areas is patchy. As one might expect, simulations reveal higher concentrations of PM2.5 in more densely populated districts. This pattern is however significantly distorted by intricate interplay of local air circulations and turbulent

diffusion in the lower parts of the atmosphere. Although there is an overall near-surface air flow towards the fjord, pollution tends to concentrate in some stagnation zones on the way. Thus, the study allows for certain generalizations. Our results reveal that the highest PM2.5 concentrations are created by long-distance horizontal air advection into stagnation zones. Gradual accumulation and convergence of the polluted air is more important than the local dispersion over emission sources. A

considerable fraction of pollution in the stagnation zones is advected from upstream urban districts. Figure 2 shows the near-surface wind- and temperature fields in the actual boundaries of the Bergen municipality for the two pre-defined weather scenarios. This area is sufficiently distant from the model boundary zone, which is shown as a grey rectangle in Figure 1 (top panel).

Our simulations represent two weather scenarios. Scenario 1 has higher water temperature (Figure 2; the top panel). It

represents the conditions with strong water-land surface temperature difference. Warmer water forces stronger wind convergence over the fjord and increases the outflow from the central valley. In some more general sense, it can be seen as a scenario with effective urban ventilation with local breeze. The increased wind convergence is also observed over smaller or medium-size water bodies. We found that these surface-layer breeze-like circulations, which are driven by the horizontal temperature differences, are to the large degree autonomous, existing independently and in many places in opposition to the

upper air winds. Scenario 2 has lower water temperature (Figure 2; the bottom panel). Weaker convergence of the near-surface winds produces different configuration of the local air flows, and therefore, different pollution pathways. This scenario describes a situation when local surface heterogeneity is not strong enough or not well organized to increase urban ventilation. Its most pronounced effect is in the formation of air stagnation in the central urban districts. Stronger upper layer winds do not break this stagnation. Thus, this stagnation zone is robust, which is an important result from the citizens and stakeholders'

point of view.

A combination of uneven distribution of emission sources with an intricate pattern of local air circulations produces a patched picture of the pollution concentrations. Our simulations identify the pathways of pollution in the surface layer. In both scenarios, the high concentrations are found not only in the areas with the highest emission rates but also in other areas where local winds advect the pollutants and concentrate them in converge zones. Figure 3 presents the map of the PM2.5

concentrations near the surface. It is simulated using the current distribution of new and older stoves. Both weather scenarios reveal high concentrations in cold air pools designating the effect and importance of topographic sheltering.

## 4.1 Implications of pollution advection across municipal districts

Now, we shall look at the role of long-distance pollution advection and convergence in different municipal districts. It is perhaps not surprizing that the highest concentrations are found in the central, densest urban districts as, e.g., just south and

upward in central Bergen valley. More detailed analysis, however, suggests that a considerable fraction of the concentration shall be attributed to advection of polluted air from other upstream districts. Bergen has eight administrative districts. Figure 4 shows the simulated near-surface concentration obtained for emission sources (stoves) that are active only in the central district (Bergenhus) and in the most populated district (Årstad). We plot both the absolute and relative concentrations. The high concentrations in Årstad are determined by the stagnant cold air pool in a local topographical shelter and the high

concentration of small apartments in old houses, each having individual wood-burning stove. In addition, the local wind convergence adds to increase the concentration even in the most polluted districts. Here, 80% to 90% of the total concentrations are of the local origin within the district itself. The non-local advection adds up to 60% to the concentrations in the most central district of Bergenhus (see Fig. 4, top panel). This district is built-up by modern high-rise buildings where stoves are not in use. Nevertheless, the PM2.5 concentrations are relatively high there. The pollution in this district is advected from the upstream

Årstad district following the circulation pathway along the northern slope of the Bergen valley. By contrast, emissions within the city centre have almost no impact on other districts as the area is located down-stream in the Bergen valley and next to the waterfront (Byfjorden). Scenarios 1 and 2 are very similar to each other with respect to the pollution transboundary transport and non-local contribution of district's emission sources. Scenario 2 leads to lower local contribution (not shown), while the effects on the other areas remain similar. Since we want to identify the conditions with severe impact on the population, we

will focus on details of the scenario 1 simulations.

Looking at the three districts located further south in the municipality, we clearly see the effect of the down-stream and -valley transport towards the city centre. The main polluter, both locally (80%) and downstream (up to 60%), is the Årstad district. The pollution trapped in the surface layer are however transported over long distances within the municipality. According to our simulations, the two southernmost districts (Fana, Ytrebygda) add up to 20% to the concentrations in the city centre, while

standing for near 100% of their own local pollution with their districts. The picture is very different when western (Fyllingsdalen, Laksevåg) and eastern (Åsane, Arna) urban districts are considered. Those districts are located behind mountains and water areas looking from the central Bergen perspective. Due to this, surface layer pollution will not be transported to the central valley, thus having no impact on the concentration of air pollutants. The local concentrations in these peripheral districts are less significant, due to different local topographic steering of the near surface air masses.

## 4.2 Assessment of mitigation actions to reduce emissions from residential wood combustion

The most environmentally friendly clean air mitigation action is to place bans on all emissions from private wood stoves. However, the need for heating during cold winter days implies that more realistic policy actions must be considered. Moreover, active use of the wood-burning stoves could be an effective retrofitting measure helping to save electricity and fossil fuels (Felius et al., 2019). Therefore, a massive transition to new stoves is deemed necessary both from political and environmental points of view. The results of our simulations impose two important constrains on plausible retrofitting measures. First, we argue and demonstrate it with model runs that there is a tight connection between weather conditions and exceedance of the concentration thresholds in particular districts. We found that PM2.5 readings in one district, say Åsane, may tell little about the concentrations in another district or even over smaller neiborhoods. So, the measures need to be constraint by physio-geographical boundaries, which in turn could be determined only through high resolution atmospheric simulations (or analysis of dense observations if they are available). Second, if the target is not to cross a certain concentration threshold, the measures must account for the developing local atmospheric circulations, and not so much for dispersion of the very local emissions.

The presented simulations and many additional sensitivity simulations suggest that unavoidable inaccuracies and uncertainties of emissions should not be overdriven in mitigation measures. Indeed, there are many uncertainties associated with the current usage and replacement of the stoves, which includes uncertainties related to the emission factor (Seljeskog et al., 2017). However, whatever high or low the specific emission factor is, the spatial redistribution and aggravation of pollutants is robustly simulated with stagnation (high concentration) zones found over the identified areas. In particular, the convergence zone related to the breeze inflow and land outflow is of general interest as it might be found in many coastal cities. Hence,

limiting economic incentives for stove replacement to just certain districts could be a cost-effective measure. We however acknowledge that other, e.g., political, factors may render such an efficiency sub-optimal. In our model simulations, we can alter the configurations and rates of emission sources in specific districts to estimate effects of different mitigation actions on the local and total level of urban pollution.

Plausible alternatives for the total and district-wise replacements of stoves with cleaner burning technology are presented in Figure 6. We change the current distribution of the older and new stoves in two clusters of four districts each. First, we replace all stoves in the Bergen valley (the Bergenhus, Årstad, Fana and Ytrebygda districts) with the new stoves only. The simulations result in strongly improved air quality with the highest concentrations being almost eliminated. Concurrently, the moderate and low concentrations are not markedly lower determining the baseline level of pollution from the new stoves – all stoves

will pollute. Then, we replace stoves in other districts (Laksevåg, Fyllingsdalen, Åsane and Arna). It results in further reduction of concentrations in the total accumulated concentrations of over the entire municipality.

We summarize the effect of mitigation actions in Figure 7 and Table 1. The national regulators set the concentration threshold of 40 µg m$^{-3}$ for PM2.5 as the delimiter of high and low air pollution. The simulations reveal that majority (respectively 17818 and 17243) of the exposed households are exposed only to low air pollution in both scenarios. Nevertheless, a total of 4031

(scenario 1) and 5986 (scenario 2) households are still exposed to high air pollution under the current distribution of stoves. Switching to the new stoves in the entire city results in elimination of high air pollution exposure. We discover that a very similar effect could be obtained with more modest but better targeted efforts focussed replacements of stoves on the central Bergen valley (Bergenhus, Årstad, Fana and Ytrebygda districts). Only 2 (scenario 1) and 44 (scenario 2) households are still exposed to high air pollution in the simulations. By contrast, 3848 and 5636 households respectively remain exposed when the

stoves are installed only in peripheral districts (Laksevåg, Fyllingsdalen, Åsane and Arna). That corresponds to the relative exposure reduction by 5% and 6% of households only. At the same time, the whole 95% of households will continue to be exposed to high air pollution of such policy action.

## 5 Discussion and conclusions

Despite the frequent depiction of residential wood burning as clean energy, urban air data clearly associate wood combustion with elevated concentrations of the particulate matter (PM2.5) in cities. These particles are extremely harmful to humans because their small size allows them to reach the lungs and to deliver toxic contaminants they adsorbed. The World Health Organization attributed 61000 of the premature deaths in 2010 alone in Europe to outdoor PM2.5 pollution from residential wood combustion; whereas in Denmark, estimations suggested that up to 30-40% to the premature deaths should be attributed

to this type of air pollution (Cincinelli et al., 2019). Residential wood combustion has its immanent importance for air quality not only in the Nordic cities where it contributes up to 82% of the total particulate matter in Northern Sweden (Krecl et al., 2008), but also elsewhere in Europe (up to 60% in Southern Germany (Bari et al., 2009)). Even in sub-tropical cities, e.g., Santiago, Chili (Mazzeo et al., 2018), impact of residential wood combustion could be critical for urban health, so that its modelling is in high demand. Due to these factors, there is a persistent interest and concentrated efforts to include the particulate

matter into an urban integrated system (UIS) modelling approach (Grimmond et al., 2020). The UIS approach combines meteorological and air quality modelling with socio-economic and policy scenarios. It has been however recognized that integration of high-resolution models in existing systems still requires further development. In this study, we described a perspective UIS approach that has emerged from our co-production efforts with local policy makers.

PALM has been proposed as a perspective model that would allow resolving urban dispersion from highly localized pollution

sources (Esau et al., 2021; Heinze et al., 2017; Maronga et al., 2019b; Zhang et al., 2020). PALM simulations were evaluated in several case studies but so far, this study is the first model application for policy and meteorological scenario development at the level of the whole administrative unit – an urban municipality. In this sense, our study sets up methodology and present perspective ways to utilize the large-eddy simulation model in the UIS. This modeling is advantageous for population-weighted exposure estimates that are still missing the high-resolution meteorological component (Savolahti et al., 2019). We demonstrate

what such a modeling component could contribute for such expose estimates (Figure 7).

The UIS studies benefit from co-production with local policy actors. Our case study in Bergen was co-designed with city managers to select the most relevant scenarios and targets. The Bergen city council has banned firing of the older stoves since

January, 1st, 2021. To facilitate transition to the new stoves, financial incentives of 5000 Norwegian krones (about 500 €) per stove have been allocated. The total allocated amount is 50 million Norwegian krones or just 25% or the amount needed to support the total replacement of the older stoves in the municipality. So, it is not surprizing that by 2021, Bergen still has more than 30 000 older stoves. Our simulations however suggest that targeted incentives might be sufficient to cover the stove replacement in the most critical urban districts. We show that such a selective replacement could have a major effect on pollution reduction below the threshold of 40 µg m$^{-3}$ of PM2.5 for almost all households in Bergen municipality. Thus, we conclude that the turbulence-resolving atmospheric modelling has a potential to contribute to the design of more efficient policy and scenarios than those presently applied. Analysis of simulation scenarios may not only increase the effects of limited incentives, which is desired per se, but can also assure that "no one is left behind" or in other words, that all households will experience positive effects of the proposed policy measures; no district would remain unnecessarily polluted or even with worse air quality.

There are available other modelling approaches and more comprehensive studies of residential wood combustion, notably, applications of the Norwegian modelling system based on the MetVed emission model, the EPISODES air quality model and AROME weather model (Grythe et al., 2019; Grythe and Lopez-Aparicio, 2020). Both the data graining (250 m) and atmospheric model resolution (1000 m) in this modelling system are much coarser than in the present study (30 m and 10 m correspondingly). Kukkonen et al. ( 2018) presented PM2.5 simulations in Helsinki, Finnland, at higher resolution (100 m) with the SILAM modelling system. This system relies on the Gaussian (down-gradient) approach and hence cannot resolve aggregation of polluted air. Emission inventory analysis and modeling results for different systems were presented in a study in four (Oslo, Helsinki, Copenhagen and Umeå) cities (Kukkonen et al., 2020). It considers the mean concentrations and the city-scale level and does not discuss relevant meteorological conditions. Our study is less model and site specific. Its focus is on explicit modelling of the turbulent dispersion and influential meteorological scenarios. Our simulations represent moderate to strong impact of the atmospheric stability on the concentration exceedance. The leading effect of atmospheric stability have been found in other our studies (Wolf et al., 2014, 2020) as well as by other authors concerned with the urban wood combustion (Grange et al., 2013). We show that this resolution and methodological differences do matter. Although the overall effect of the stove replacement programmes on the emission reduction in the city could be moderate, the replacement in certain sensitive

districts helps to dramatically reduce the population expose. In this sense, our study supports the conclusions of Lopez-Aparicio

and Grythe ( 2020) study in Oslo, Norway, that the current indiscriminate subsidy programmes are ineffective.

Concluding the modelling study of urban pollution dispersion, we observe that local winds within the turbulent boundary layer can transport polluted air over long distances in the urban areas. Convergence and accumulation of pollution from distributed small-scale emission sources, such as the considered household stoves, deteriorate air quality in stagnation zones where the local emissions are not sufficient to maintain such high concentrations. We simulated only the worst-case scenarios for

wintertime air pollution, which are associated with the coldest weather and most frequent use of wood stoves for heating. Therefore, our concentrations considerably exceed the mean values given by other studies (Grythe et al., 2019; Kukkonen et al., 2020). Other weather regimes produce less severe air pollution which might however contribute over the whole heating season. The weather conditions favourable for air quality deterioration in Bergen are considered in other our works (Wolf et al., 2020; Wolf and Esau, 2014). Two studies by Wolf et al. ( 2017b, 2014) specifically consider co-variability between the

temperature inversions and weather regimes in Bergen as the main control factors of the local atmospheric pollution. Emission scenario testing like the one highlighted here can be an important step of such a reasoning.

The methodology of this study is experimental. It requires further development and integration with other physical and chemical components of the urban environment. Such efforts are on the way (Maronga et al., 2019b). More specifically, the quality of the PALM simulations needs assessment against in situ observations of concentrations as well as against more

constraining laboratory and other model results (Gronemeier et al., 2017, 2021). The input data sets ought to be more constrained too (Heldens et al., 2020; Masson et al., 2020). Uncertainty in emissions and surface boundary conditions currently impede fine tuning of the simulations. Societal uncertainties could be also influential (Felius et al., 2019; Wyss et al., 2016). Heating patterns and habits are poorly constrained. Concerns that the new stoves will be used more, and therefore retard the effect on air pollution, have been also raised by stakeholders. A broader study of societal effects is out of scope of this work.

Finally, the PALM model is under extensive development now to allow for more realistic and comprehensive urban meteorological studies. It will include a block for improved surface boundary conditions (Maronga et al., 2019a, 2019b; Resler et al., 2017) and connection with the regional model COSMO (Fröhlich and Matzarakis, 2020).

**Code availability**

The model code PALM is available from the PALM group from https://palm.muk.uni-hannover.de/trac/wiki/doc/install. The

model is free to download and to use upon registration. The scripts for data analysis and visualization are written in MATLAB

and available from the Nansen Center ftp server at ftp://ftp.nersc.no/igor/ .

**Data availability**

Availability conditions for the collection of data sets to setup and run the model and weather scenarios are described in Wolf

et al. (2020). A selection of modelling results and processed data (smaller data files) are available from the Nansen Center ftp

server at ftp://ftp.nersc.no/igor/ . Data from new model simulations (larger data files) for this study are available upon request

to the corresponding author.

**Author contributions**

Tobias Wolf was responsible for the data acquisition, setting and running the model PALM, data analysis scripts and

visualization routines. He contributed to writing the manuscript and interpretation of the modelling results. Lasse Pettersson

was responsible for contacts with data providers and end-users, design of scenarios, interpretation, and communication of the

results. He contributed to writing the manuscript. Igor Esau was responsible to the conceptual design of the study, methodology

of numerical simulations and data analysis. He wrote the manuscript and connected the results of the study with the need of

end-users.

**Competing interests**

The authors declare that they have no conflict of interests.

**Special issue statement**

This study is a contribution to a special issue "Pan-Eurasian Experiment (PEEX) – Part II" of the Atmospheric Chemistry and

Physics journal.

## Acknowledgements

This study is performed within the GC Rieber Climate Research Institute at the Nansen Center. Our studies of air pollution in the Bergan have capitalized on stakeholder and user perspectives from and cooperation with Ulrik Jørgensen, Sverre Østvold, Nils Møllerup and Even Husby – Port of Bergen, and Eva Britt Isager and Per Vikse – Climate section, Mette Iversen and Nils-Eino Langhelle – Section for Plan and Geodata, Per Hallstein Fauske and Arve Bang, Health Care Agency, all Bergen municipality.


## Financial support.

This research has been supported by the GC Rieber Foundations, Bergen, and strategic institute funding (RCN grant no. 218857).

The finalization of data analysis and publication were supported by the Norway Grant TO01000219 (TURBAN). The project
"TURBAN - Turbulent-resolving urban modeling of air quality and thermal comfort" benefits from a research grant from Norway and Technology Agency Czech Republic (TA CR).

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

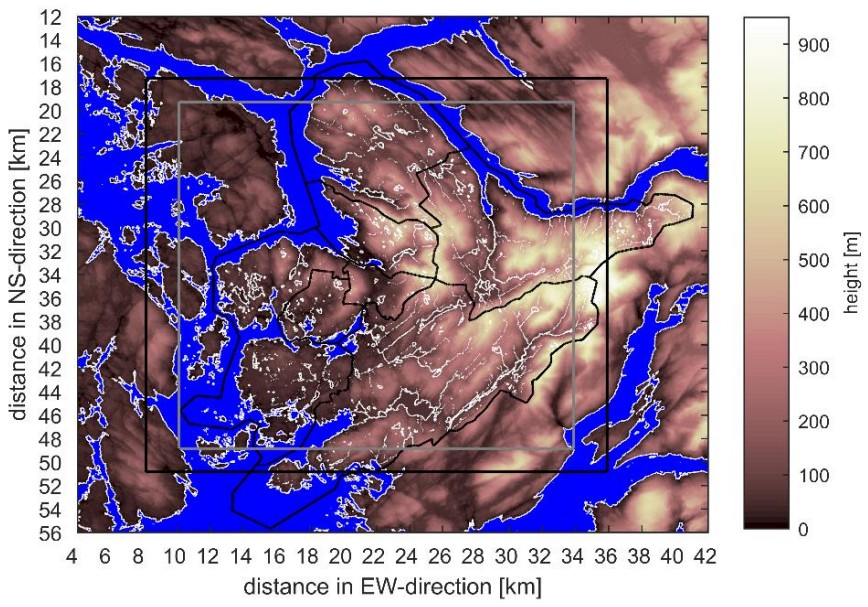

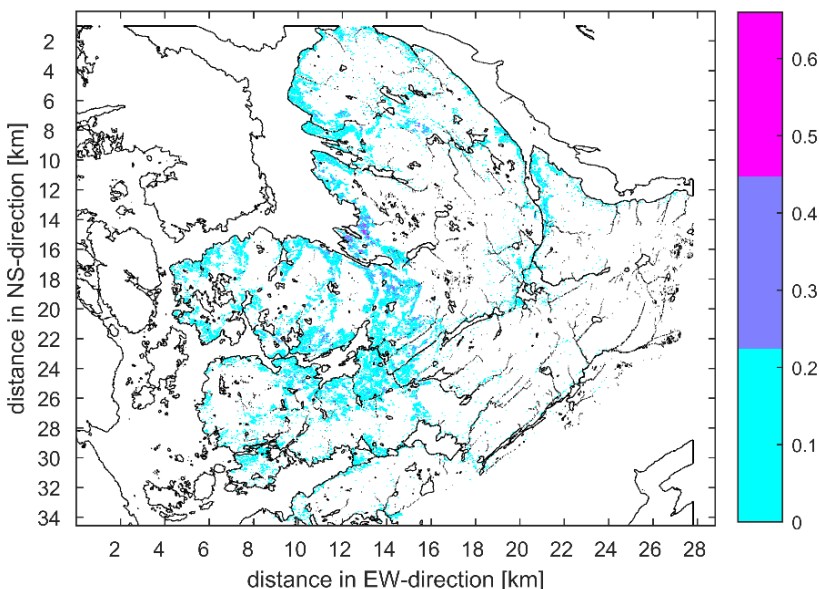

Figure 1: *Top*: Topographic map of the Bergen municipality. The colour shading indicates elevations of the land surface area. Water surfaces are marked with blue colour shading. The black curved lines indicate district boundaries. The black rectangle indicates the full extent of the model simulation domain. Buffer zones are not included. The grey rectangle indicates the part of the model domain that is selected for the analysis. *Bottom*: Map with the locations of wood-burning stoves in Bergen. The colour shading indicates the number of households with registered stoves per 10 x 10 $m^2$ grid box (averaged over 3 x 3 boxes). Black lines indicate waterfront. Data sources are the Bergen Municipality and the Bergen Fire department.

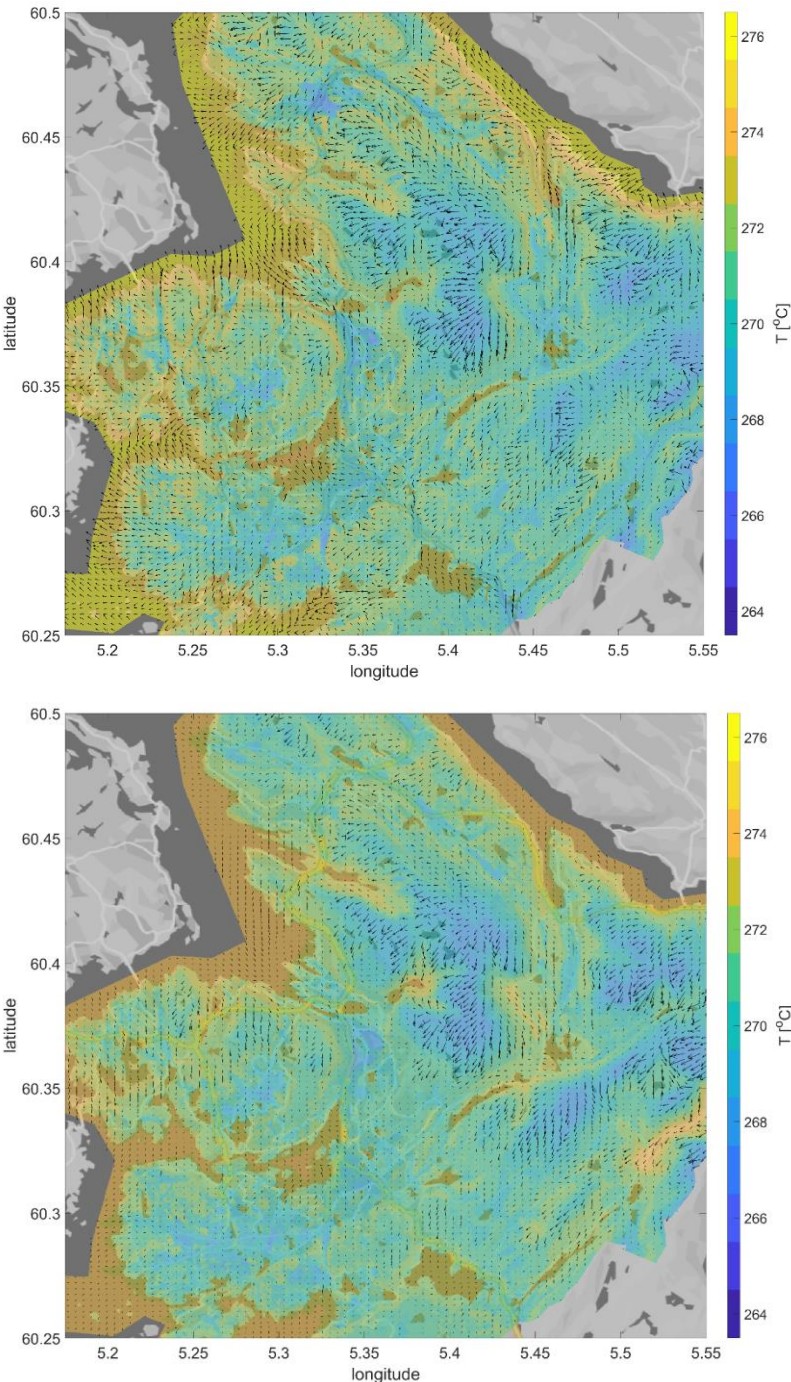

**Figure 2: Surface layer meteorology obtained from the PALM model runs for the scenario 1 (top) and 2 (bottom). The colour shading represents the surface air temperature at 2 m above the ground (terrain-following temperature field). Vectors represent the surface-layer wind at 10 m. Data overlay the grayscale map from Map data © 2019 Google.**


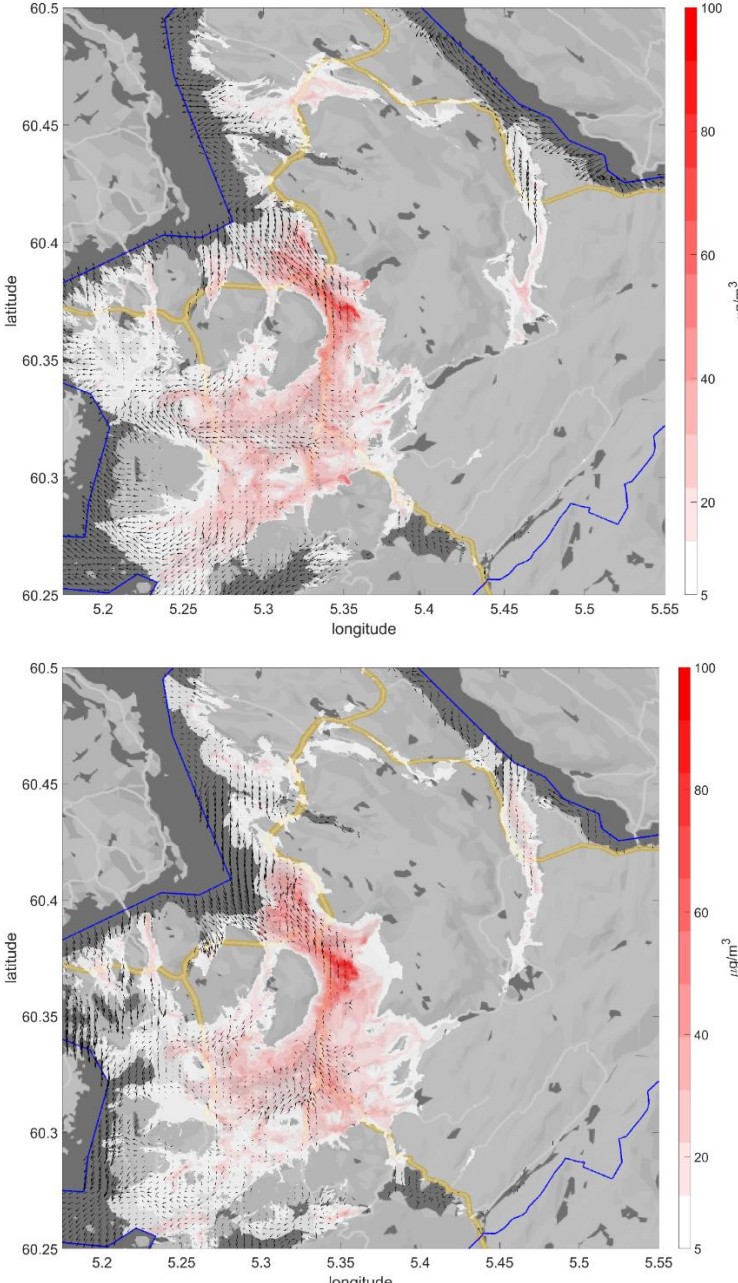

**Figure 3: Simulated concentrations at 5 m above the ground (the surface layer concentrations) of PM$_{2.5}$ for the scenarios 1 (top) and 2 (bottom). Vectors show the surface wind at 10 m. The simulations use the current distributions of the wood-burning stoves and the current fraction of the new and older stoves in Bergen. Data overlay the grayscale map from Map data ©2019 Google.**


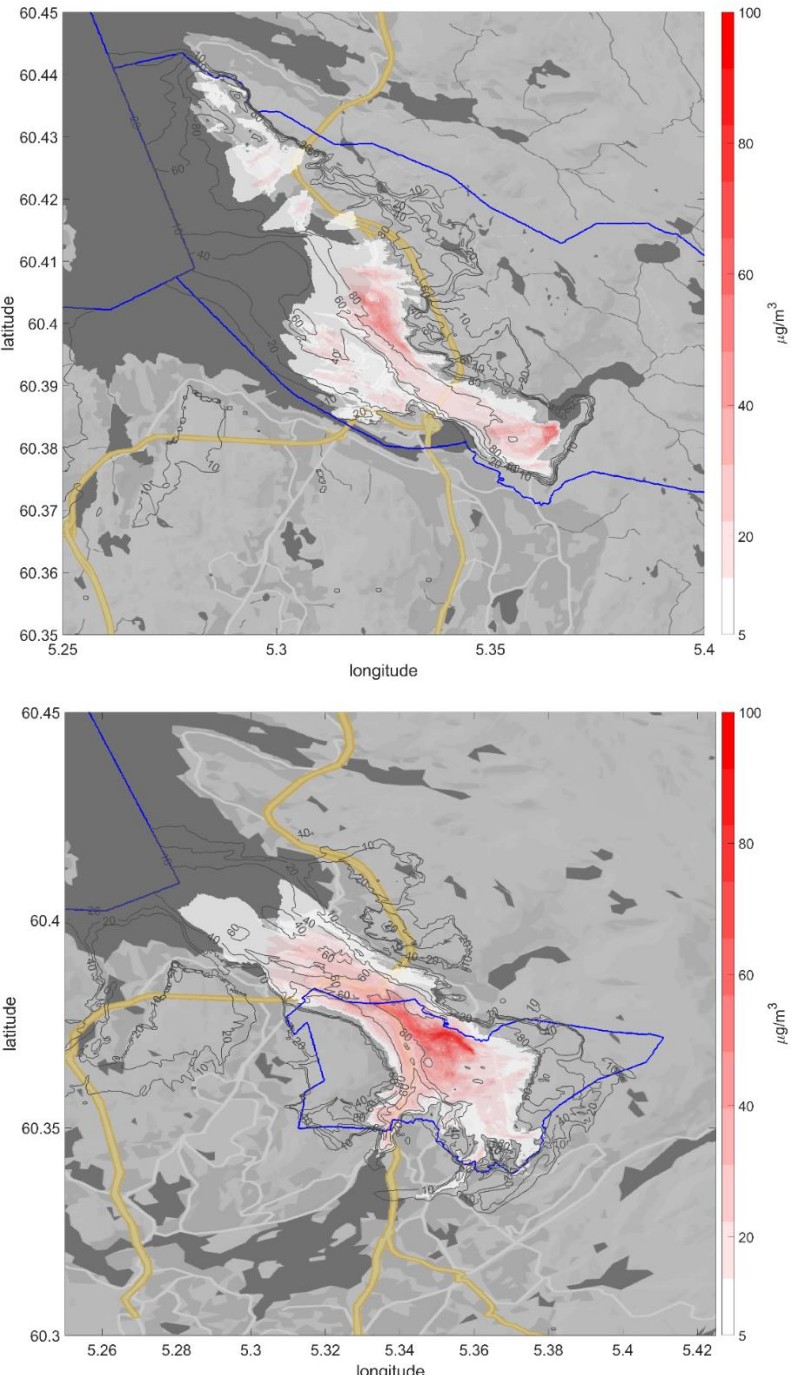

**Figure 4: The simulated surface-layer concentrations of PM$_{2.5}$ for the scenario 1 (shown in Fig. 3, top panel) but for emissions only from stoves within the Bergenhus (top) and Årstad (bottom) districts. The simulations use the current distributions of the number and types of stoves. The blue lines show the district boundaries. Thin grey isolines indicate the fraction of the total pollution that was caused by the emissions within the respective district. Data overlay the grayscale map from Map data ©2019 Google.**

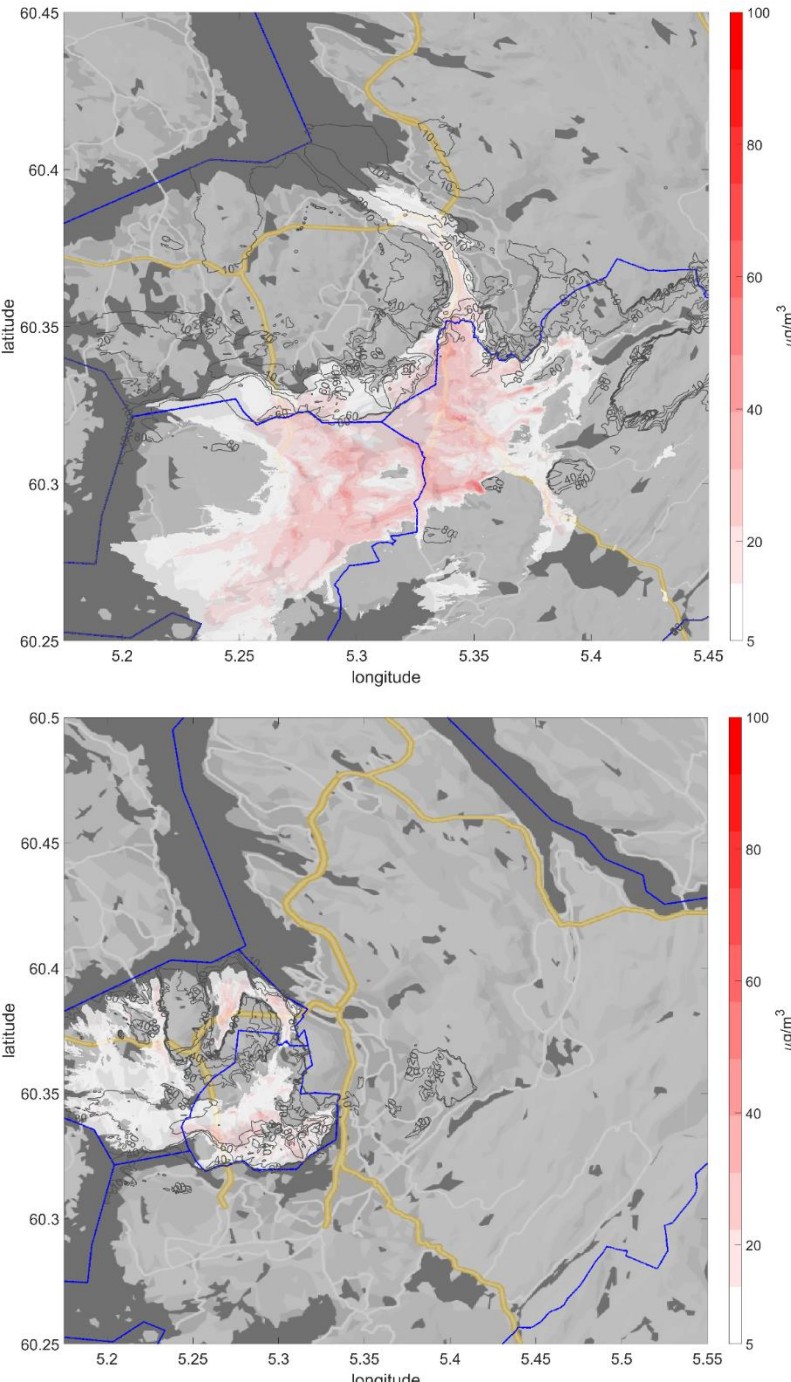

Figure 5: Same as Figure 4 but for the emissions within the districts Fana and Ytregygda (top), and Laksevåg and Fyllingsdalen (bottom). Data overlay the grayscale map from Map data ©2019 Google.

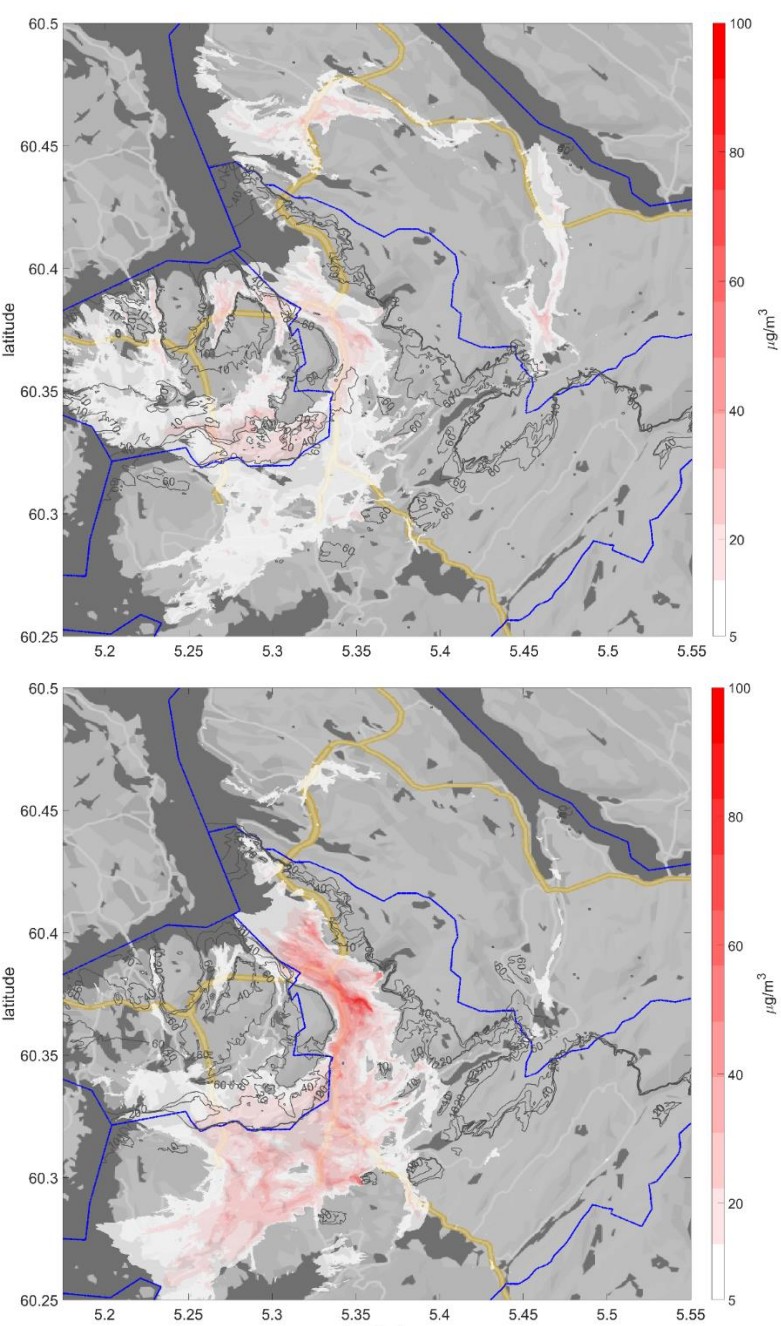

Figure 6: The same as in Figure 4 but for the emissions expected when all older stoves have been replaced with the new stoves in Bergenhus, Årstad, Fana and Ytrebygda districts (top) or in Laksevåg, Fyllingsdalen, Åsane and Arna districts (bottom). The boundaries of the districts are highlighted with the blue lines. Thin grey isolines indicate the fraction of the total pollution that was caused by the respective districts, when compared to the total pollution distribution in Figure 3 (top). Data overlay the grayscale map from Map data ©2019 Google.


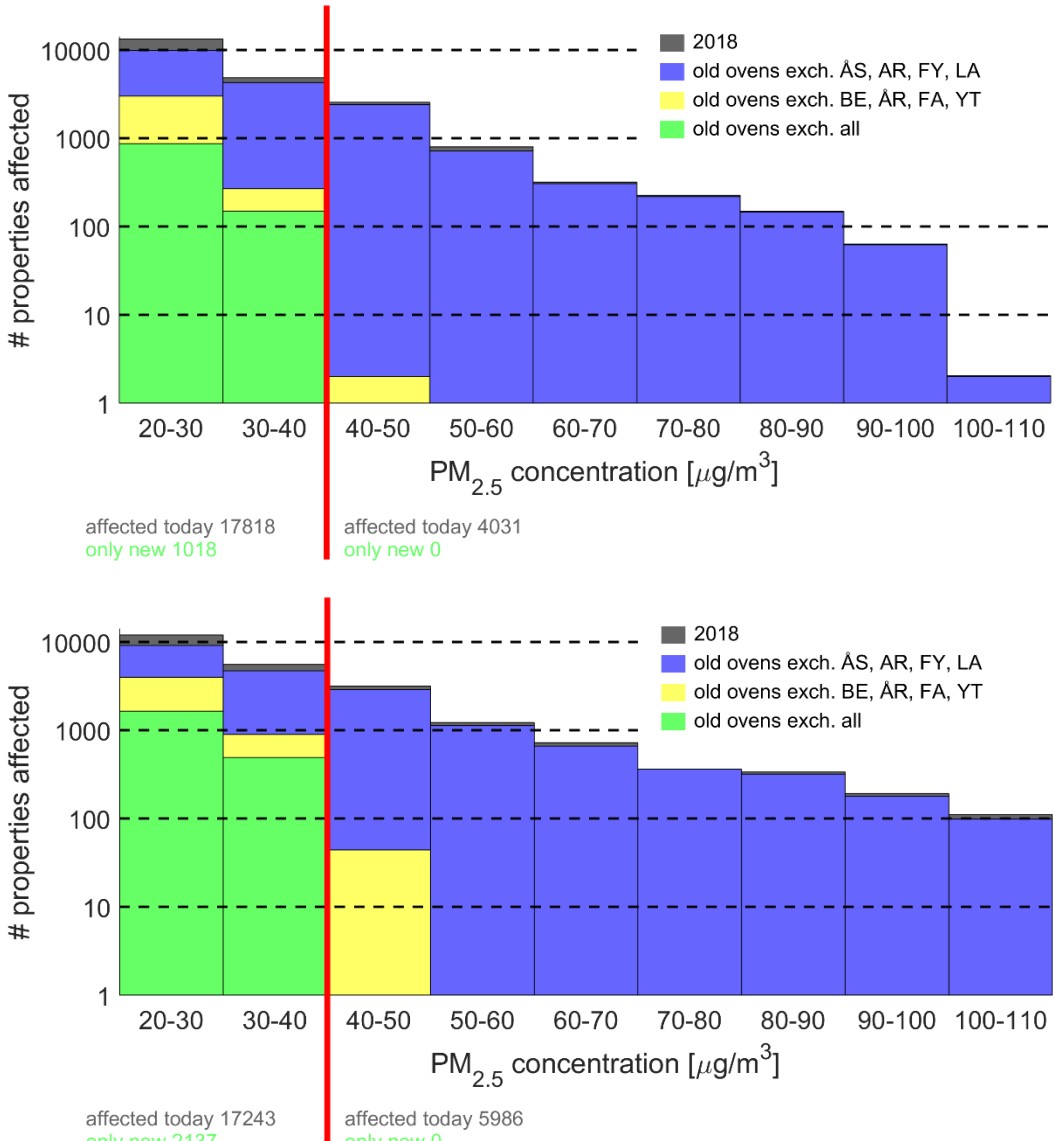

**Figure 7:** Histogram of the accumulated number of households (# properties on the y-axis given in logarithmic scale) in Bergen that exposed for different concentrations (the binned concentration intervals on the x-axis) for the scenario 1 (top) and 2 (bottom). Abbreviations are: "2018" (upper part of all columns) means the exposure from the simulations with the current mix of older and new stoves; "old stoves exch." means a complete replacement with the new stoves in various subsets of districts (as shown in Figure 6). The impact of the complete replacement in Åsane (ÅS), Arna (AR), Fyllingsdalen (FY) and Laksevåg (LA) (Figure 6, bottom) is shown as blue columns; the same in Bergenhus (BE), Årstad (ÅR), Fana (FA) and Ytrebygda (YT) (Figure 6, top) is shown as yellow columns; the same in all districts – as green columns. The threshold of 40 μg m$^{-3}$ for PM$_{2.5}$ concentrations is indicated with a vertical red line.

**Table 1. Summary of the total simulated effect of the suggested mitigation actions.**

| | Weather scenario 1: Winter temperature inversion | | Weather scenario 2: Winter temperature inversion and stagnation | |
|---|---|---|---|---|
| Air pollution level | Medium | Strong | Medium | Strong |
| PM2.5 concentration range ($\mu$g m$^{-3}$) | 20 - 40 | 40 - 110 | 20 - 40 | 40 - 110 |
| **The present-day situation:** the total number of exposed households | 17 818 | 4 031 | 17 243 | 5 986 |
| Reduction in the number of the exposed households when transition to the new stoves is completed in Åsane, Arna, Fyllingsdalen and Laksevåg districts | 20% | 5% | 20% | 6% |
| Reduction in the number of the exposed households when transition to the new stoves is completed in Bergenhus, Årstad, Fana and Ytrebygda districts | 82% | 100% | 72% | 99% |
| Reduction in the number of the exposed households when transition to the new stoves is completed in the whole Bergen municipality | 94% | 100% | 88% | 100% |