# Peer review of "Dispersion of particulate matter $(PM_{2.5})$ from wood combustion for residential heating: Optimisation of mitigation actions based on large-eddy simulations"

_Atmospheric Chemistry and Physics, 2021_

## Author Comment (AC1)

**Reply to the comments to the discussion of the preprint.**

**My general comments were provided already in the previous review phase:**

*The research questions and methods are novel. However, the biggest issue is that, at its current state, the manuscript does not fulfil one of the main aims of ACP: "The journal scope is focused on studies with general implications for atmospheric science rather than investigations that are primarily of local or technical interest." Currently, the manuscript is very Bergen specific and resembles a project report to the municipality. Hence, for the manuscript to fit into the scope of ACP, a major review is needed.*

We are glad to read about the reviewer appreciation of our study. It encourages us to push harder in advancing research in this direction. We also agree that the case study in the manuscript is perhaps too extensive and too specific. However, we think that our arguments are also worth to consider. We want to advance the numerical modeling technique to realistic applications. How would one achieve that without reporting success and failure with a specific case? We all rely on peer-review process to correct and solidify the research. Understandably, the Bergen municipality does not have proper competence to evaluate the work against the research standards. So, we are interested in open and honest discussion of the case study as well, not only the general part of the study. Besides this point, it was and still is rather routinely accepted in the community journals, including the ACP journal, to publish insides into seemingly specific but by the mater of fact useful case studies. Let us look at ACP publications. Among the most downloaded articles you may find the _following case specific studies_:

93,433 downloads: Christoudias, T. and Lelieveld, J.: Modelling the global atmospheric transport and deposition of radionuclides from the Fukushima Dai-ichi nuclear accident, Atmos. Chem. Phys., 13, 1425–1438, https://doi.org/10.5194/acp-13-1425-2013, 2013.

63,272 downloads – very close to our subject of study - Benton, A. K., Langridge, J. M., Ball, S. M., Bloss, W. J., Dall'Osto, M., Nemitz, E., Harrison, R. M., and Jones, R. L.: Night-time chemistry above London: measurements of NO3 and N2O5 from the BT Tower, Atmos. Chem. Phys., 10, 9781–9795, https://doi.org/10.5194/acp-10-9781-2010, 2010.

31,941 downloads – very close to our subject of study - Zhang, R., Jing, J., Tao, J., Hsu, S.-C., Wang, G., Cao, J., Lee, C. S. L., Zhu, L., Chen, Z., Zhao, Y., and Shen, Z.: Chemical characterization and source apportionment of PM2.5 in Beijing: seasonal perspective, Atmos. Chem. Phys., 13, 7053–7074, https://doi.org/10.5194/acp-13-7053-2013, 2013.

and among the most recent such publications (1,900 downloads): Hellén, H., Kangas, L., Kousa, A., Vestenius, M., Teinilä, K., Karppinen, A., Kukkonen, J., and Niemi, J. V.: Evaluation of the impact of wood combustion on benzo[a]pyrene (BaP) concentrations; ambient measurements and dispersion modeling in Helsinki, Finland, Atmos. Chem. Phys., 17, 3475–3487, https://doi.org/10.5194/acp-17-3475-2017, 2017.

Those arguments support out conclusion that our study is well fitted to the ACP journal and will be interesting for the larger community of readers. In any case, however, we would suggest leaving the final decision on the topical editor. We hope that the reviewer will agree with that.

*Some general comments:*

*1. As mentioned above, the manuscript is now only focused on the city of Bergen and hence the results are lacking general implications. For instance, no comparison with previous studies applying more simplified geometries or real topographies is given. Furthermore, there is rather a lot of discussion about the funding of these kinds of studies by cities, which I think does not fit the scope of ACP.*

We hope that we answered to this concern above. As for previous studies, there are no such studies directly comparable with our results either by method or by subset of input data.

As for "the funding discussion", we believe that this is a matter of certain misunderstanding. We do not discuss funding of studies in the published preprint. We discuss how our modeling methodology might help to optimize socio-economic policy scenarios. To our view, this is important aspect of the science published by ACP, just look at the list of the most influential relevant papers given above.

*2. Now the manuscript is difficult to follow. This is partly related to the language and partly to the structure of the manuscript. At least these points require improvements:*

*- The aims of the study must be stated clearly*

*- Sections 1 & 2 should be merged because they overlap a lot regarding the content.*

*- The language requires revision. Firstly, the paragraphs are lacking coherence and the text is missing flow. Secondly, the application of articles (a/the) and prepositions must be double-checked.*

The required corrections have been introduced. We disagree that the paper is difficult to follow. It is written in plain language checked by the English speaker. The structure complies with the IMRAD standard for research papers. We will provide further revision in the final version of the manuscript if any.

Thank you for your minor but very important corrections. We have now included all of them into the text.

---

## Author Comment (AC4)

**Reply to the comments to the discussion of the preprint.**

**My general comments were provided already in the previous review phase:**

*The research questions and methods are novel. However, the biggest issue is that, at its current state, the manuscript does not fulfil one of the main aims of ACP: "The journal scope is focused on studies with general implications for atmospheric science rather than investigations that are primarily of local or technical interest." Currently, the manuscript is very Bergen specific and resembles a project report to the municipality. Hence, for the manuscript to fit into the scope of ACP, a major review is needed.*

We are glad to read about the reviewer appreciation of our study. It encourages us to push harder in advancing research in this direction. We also agree that the case study in the manuscript is perhaps too extensive and too specific. However, we think that our arguments are also worth to consider. We want to advance the numerical modeling technique to realistic applications. How would one achieve that without reporting success and failure with a specific case? We all rely on peer-review process to correct and solidify the research. Understandably, the Bergen municipality does not have proper competence to evaluate the work against the research standards. So, we are interested in open and honest discussion of the case study as well, not only the general part of the study. Besides this point, it was and still is rather routinely accepted in the community journals, including the ACP journal, to publish insides into seemingly specific but by the mater of fact useful case studies. Let us look at ACP publications. Among the most downloaded articles you may find the _following case specific studies_:

93,433 downloads: Christoudias, T. and Lelieveld, J.: Modelling the global atmospheric transport and deposition of radionuclides from the Fukushima Dai-ichi nuclear accident, Atmos. Chem. Phys., 13, 1425–1438, https://doi.org/10.5194/acp-13-1425-2013, 2013.

63,272 downloads – very close to our subject of study - Benton, A. K., Langridge, J. M., Ball, S. M., Bloss, W. J., Dall'Osto, M., Nemitz, E., Harrison, R. M., and Jones, R. L.: Night-time chemistry above London: measurements of NO3 and N2O5 from the BT Tower, Atmos. Chem. Phys., 10, 9781–9795, https://doi.org/10.5194/acp-10-9781-2010, 2010.

31,941 downloads – very close to our subject of study - Zhang, R., Jing, J., Tao, J., Hsu, S.-C., Wang, G., Cao, J., Lee, C. S. L., Zhu, L., Chen, Z., Zhao, Y., and Shen, Z.: Chemical characterization and source apportionment of PM2.5 in Beijing: seasonal perspective, Atmos. Chem. Phys., 13, 7053–7074, https://doi.org/10.5194/acp-13-7053-2013, 2013.

and among the most recent such publications (1,900 downloads): Hellén, H., Kangas, L., Kousa, A., Vestenius, M., Teinilä, K., Karppinen, A., Kukkonen, J., and Niemi, J. V.: Evaluation of the impact of wood combustion on benzo[a]pyrene (BaP) concentrations; ambient measurements and dispersion modeling in Helsinki, Finland, Atmos. Chem. Phys., 17, 3475–3487, https://doi.org/10.5194/acp-17-3475-2017, 2017.

Those arguments support out conclusion that our study is well fitted to the ACP journal and will be interesting for the larger community of readers. In any case, however, we would suggest leaving the final decision on the topical editor. We hope that the reviewer will agree with that.

*Some general comments:*

*1. As mentioned above, the manuscript is now only focused on the city of Bergen and hence the results are lacking general implications. For instance, no comparison with previous studies applying more simplified geometries or real topographies is given. Furthermore, there is rather a lot of discussion about the funding of these kinds of studies by cities, which I think does not fit the scope of ACP.*

We hope that we answered to this concern above. As for previous studies, there are no such studies directly comparable with our results either by method or by subset of input data.

As for "the funding discussion", we believe that this is a matter of certain misunderstanding. We do not discuss funding of studies in the published preprint. We discuss how our modeling methodology might help to optimize socio-economic policy scenarios. To our view, this is important aspect of the science published by ACP, just look at the list of the most influential relevant papers given above.

*2. Now the manuscript is difficult to follow. This is partly related to the language and partly to the structure of the manuscript. At least these points require improvements:*

*- The aims of the study must be stated clearly*

*- Sections 1 & 2 should be merged because they overlap a lot regarding the content.*

*- The language requires revision. Firstly, the paragraphs are lacking coherence and the text is missing flow. Secondly, the application of articles (a/the) and prepositions must be double-checked.*

The required corrections have been introduced. We disagree that the paper is difficult to follow. It is written in plain language checked by the English speaker. The structure complies with the IMRAD standard for research papers. We will provide further revision in the final version of the manuscript if any.

Thank you for your minor but very important corrections. We have now included all of them into the text.

*I hope these comments will be answered in this review phase. Another general comment:*

*- What boundary conditions are applied for the passive PM2.5?*

We specified the PM2.5 boundary conditions in more details now. The lateral and upper boundary conditions are non-periodic. It means that the substance leaving the domain does not recycled on the other side. The bottom boundary conditions are non-penetrative.

**Specific comments (P=page, L=line):**

**P1 L7: "emission" --> "emissions"**

It is changed now.

**P1 L17-18: I would leave this definition of LES out of the abstract**

We removed this sentence.

*P1 L20-21: "with the worst air pollution" --> "that typically lead to the weakest air quality"?*

We do not agree that "the weakest air quality" will improve the text. But we reformulated the sentence as "Such complex geography is expected to favour local air quality hazards, which makes this study of general interest. "

*P1 L21: "Bergen" --> "Bergen, Norway"*

It is changed now.

**P1 L21: "True laser" sounds wrong. I understand that you are meaning "topography from laser scanning" here**

It is changed now in the text to "The topographic data for the approximation are taken from a laser-scan digital elevation model (DEM) of 1 m resolution provided by the Norwegian mapping authority (Statens Kartverk, 2018).", and removed from the Abstract.

**P2 L22: "at the regular mesh" is unnecessary detailed here**

We removed this specification.

*P2 L28: "limited incentives" --> "limitation incentives"? "limited incentives" --> "limitation incentives"?*

We disagree. The proposed changes would alter the meaning.

**P2 L38: "in short run" --> "in the short run"**

We disagree. Our use of language is correct.

**P3 L46 – P4 L79: This paragraph is long and difficult to follow. You could split it up into two or more parts.**

We agree. It was a too long text. We made two paragraphs instead.

**P3 L47-48: "with the meteorological background set up by shifting weather" sounds peculiar. How about: "with the temporally varying prevailing meteorological conditions"?**

We agree. We removed the sentence.

**P3 L48: I would move "e.g. in Bergen, Norway" to the end of this phrase: "... stagnation zones, as shown in Bergen, Norway"**

Perhaps, but we did not find it optimal.

**P3 L51: "(Chandler, 1976), (Bai, 2018)." --> "(Chandler, 1976; Bai, 2018)."**

This is a typesetting issue for technical editing.

**P3 L55: "That is likely true". I would not be so sure. Low-cost sensors are not the most reliable data sources.**

We do not understand this comment. In this sentence we are talking about dense networks of sensors. Each sensor could be less reliable, but when there are hundreds of them, statistical methods are able to recover reliable information.

**P3 L58: "Locked within silo". What does this mean?**

"Silo" or "siloed structure" is a standard term from social science, which describes such an organization where information flow is allowed only between directly subordinated units, but not between units operating at the same level of hierarchy. In its application to meteorological information, it allows only for use of information/models approved by the higher levels. Any information collected around, whatever relevant or reliable it is, won't be taken up and considered in such an organization.

**P3 L60-61: I do not think that information about project funding belongs here.**

We removed this information.

**P4 L75: "massive-parallel"? You mean "supercomputers that can be applied to run massively parallelized simulations"?**

We disagree, our use of computing terminology is correct.

**P4 L83: "by Wolf et al. (Wolf-Grosse et al., 2017a; Wolf et al. 2020)" --> "by Wolf-Grosse**

**et al. (2017a) and Wolf et al. (2020)"**

This is a typesetting issue for technical editing.

**Section 2: A general figure of the area of interest (i.e., Bergen) and its districts would be useful.**

We do not understand this comment. Figure 1 is such a figure of the area of interest.

**P5 L 104: "Bergen has clean air brought to the city with westerlies from the Atlantic Ocean" sounds wrong. Why not simply: "the prevailing westerly wind provides clean from the Atlantic Ocean to Bergen"?**

We agree. The sentence is now "As in many Nordic cities, westerly winds bring clean air into Bergen from the Atlantic Ocean."

**P5 L113: "There is a strong anti-correlation between air quality and air temperature in Bergen (Wolf and Esau, 2014)." This is not Bergen-specific but applies to maybe most of the cities?**

This is not true. Air quality in temperate and tropical many cities positively correlates with air temperature. See e.g., Ramsey, N. R., Klein, P. M., & Moore, B. (2014). The impact of meteorological parameters on urban air quality. *Atmospheric Environment*, **86**, 58–67.

Moreover, Chudnovsky et al. (2014) used MODIS data product to show that the corellations change sign when temperature cross a threshold of about 7°C.

Chudnovsky, A., Lyapustin, A., Wang, Y., Tang, C., Schwartz, J., & Koutrakis, P. (2014). High resolution aerosol data from MODIS satellite for urban air quality studies. *Central European Journal of Geosciences*, **6**(1), 17–26.

A study in Seoul (seo et al., 2018) revealed positive correlations between PM10 and temperature for short term pollution episodes.

Seo, J., Park, D.-S. R., Kim, J. Y., Youn, D., Lim, Y. Bin, & Kim, Y. (2018). Effects of meteorology and emissions on urban air quality: a quantitative statistical approach to long-term records (1999–2016) in Seoul, South Korea. *Atmospheric Chemistry and Physics*, **18**(21), 16121–16137.

Yet another study in Nagasaki (Wang et al 2015) clearly revealed positive correlations between PM2.5 and temperature in all months.

Wang, J., & Ogawa, S. (2015). Effects of meteorological conditions on PM2.5 concentrations in Nagasaki, Japan. *International Journal of Environmental Research and Public Health*, **12**(8), 9089–9101.

**P5 L 114: "calm weather periods" --> high-pressure systems leading to weak winds?**

We prefer to keep our formulation.

**P6 L141: "PALM resolves" --> "LES resolves"**

We are working with a specific code PALM. There is no need to generalize here.

**P6 L141-142: "PALM explicitly resolves a part of relevant three-dimensional atmospheric turbulence dynamics as well as turbulence". I do not understand the meaning of this phrase. Yes, LES directly resolves the turbulence structures that are larger than the grid and parametrises the rest.**

Yes, we tell this to our reader.

**P7 L150: "runs"**

It is changed now.

**P7 L 156: "chemical processes". Aerosol dynamics can also have an impact.**

We agree.

**P7 L 164: Open "NO2"**

Sorry, but we did not understand this comment.

**P7 L167: "The domain includes buffer zones used for linear interpolation between the opposing period boundaries of 1000 m width". The meaning of this phrase is not clear to me.**

The sentence was replaced with "Therefore, this domain includes buffer zones with a width of 1000 m each, which are needed for linear interpolation between the opposing periodic boundaries."

**P8 L169: "the largest achieved urban simulations so far". Are you sure?**

Yes, we are still sure.

**P8 L179-184: Are the temperatures applied just some generic values or have you taken them from measurements or model simulations?**

The temperatures were taken from analysis of direct observations during the weather episodes corresponding to the simulated scenarios.

**P8 L191-192: "We have already identified the typical meteorological conditions that correspond to the high urban pollution episodes (Wolf et al., 2014)." --> I would write "The typical meteorological conditions that correspond to the high urban pollution episodes in Bergen have been identified in a previous study by Wolf et al. (2014)"**

Since journals recommend using active voice, we prefer to keep our sentence.

**P9 L194-196: These side comments about the boundary conditions applied for PALM are a bit confusing.**

We changed this comment to "This setup reflects conditions of winter temperature inversions with

radiative surface cooling over land and warmer sea surface temperatures."

**P9 L215: "Since 1995 installation only the new ovens are allowed". Something missing here?**

We changed it to "Since 1995, only new stoves are allowed."

**Section 4: please refer more to the figures. I will make it easier to follow the text.**

Thank you. We tried to follow this advice now.

**P11 L252: "the actual local conditions". Which conditions?**

We changed this to "This pattern is however significantly distorted by intricate interplay of local air circulations and turbulent diffusion in the lower parts of the atmosphere."

**P11 L253: "in more densely populated districts". This can be expected only if the emissions correlate with the population density.**

This is a good point. Indeed, districts with more high-rise buildings, and hence population, are not necessarily those with the largest emission. However, in Bergen, population density does not vary significantly at the district level. High-rise buildings are dispersed and embedded into the areas of low-rise houses.

**P11 L255: "in some stagnation zones". It would be easier to follow the manuscript if you indicated these areas in the respective figures.**

We agree, this is one of possible ways to organize the figures. Other approaches are possible as well. We tried to reduce use of local toponymics to focus on more general applicability of the study.

**P11 L262: "e.g., over Nordåsvannet at 60.32 N and 5.32 E." You could mark this in the respective figure.**

We will do this on high resolution Figure.

**P12 L281-282: "Artificially defining emissions…" This belongs to the methods section.**

We agree. These sentences were moved to the Methods section.

**Figures 4-6: I find it difficult to distinguish different districts on the figures. How about 1) combining the figures to one figure and always showing the same area in the map OR 2) adding a smaller map to each figure to show the location of the specific district on the big map?**

We tried that and several other methods, but it makes Figures overloaded with geographical information. Perhaps, a reservation of the whole page for an infographics-like illustration would help, but we are not experts in the graphical design. So far, we decided to keep figures as they are.

**P14 L329: a dot missing after "Table 1"**

We changed this.

**P14 L330: is the limit 40 ug/m3 for the daily average or temporal values?**

This is a limit for hourly averaged values.

**P14 L332: "households are under the current distribution of ovens exposed to high air pollution." The word order is confusing.**

We agree. We changed the sentence to "Nevertheless, a total of 4031 (scenario 1) and 5986 (scenario 2) households are still exposed to high air pollution under the current distribution of stoves."

**P15 L362: "Felius et al. study (Felius et al., 2019) however" --> "Felius et al. (2019), however,"**

This is a typesetting issue for technical editing.

**P16 L385: I guess the user-code modifications should be provided as well?**

We agree. The modifications are described and provided.

---

## Author Response (AR1)

**Final reply to ACP wood-burning**

**General comments:**
**The study aims to use a LES model to identify the impacts of mitigating wood stove particulate emissions on air quality over Bergen. The study can have interesting scientific and policy-related implications. However, the current version lacks a proposer discussion and analyses of the mitigation impacts and the possible extrapolation of the results in other regions as it is limited to Bergen only. Additionally, although references to previous studies are provided with respect to model description, it is still necessary to include some features of the model regarding fx other emission sources, initial and boundary conditions, and meteorological drivers. Finally, it would be good to compare the simulated PM2.5 levels with observations to better discuss the contribution of wood stoves and their mitigation. I would also recommend revision in the language as there are some section that are difficult to follow. Given the above concerns, I still encourage publication in ACP.**

*We thank the reviewer for very good indication of weak parts of our study. We now extend the discussion with respect to intercomparison to over wood-burning stove and emission studies in a broader global context. We also went through the modeling section and included more details here, specifically we explain the boundary conditions for PM2.5 and connection between PALM conditions and meteorological drivers.*

*Finally, we compare the simulated and observed PM2.5 levels for scenario conditions. More accurate intercomparison and analysis is however a subject for future (more technical) publications.*

**Specific comments:**
**Abstract:**
**Lines 20-22: Do you mean that the observed levels were higher than simulated levels?**
**This then cannot only be attributed to long-range transport as there are also biases in simulated local levels?**

*Indeed, this sentence was ambiguous. We are talking only about model results. It is now changed to:*

*"The simulated concentrations were larger than the concentrations obtained only due to the local PM2.5 emission …"*

**Section 3.1:**
**How about other sources of pollution such as traffic and other residential combustion sources? Are they (and how) treated?**
**What kind of meteorological information is used to drive the transport? These might be described in earlier publications but needs to be described briefly here for context.**

*We agree that this information is very important. Very detailed description with respect to meteorological conditions and diverse pollution sources and species has been published in*

*Wolf, T., Pettersson, L. H., & Esau, I. (2020). A very high-resolution assessment and modelling of urban air quality. Atmospheric Chemistry and Physics, **20**(2), 625–647.*

*We do not think that this paper is a good place to repeat this discussion again. We deliberately not to include other pollutants and other types of sources to look at the wood-burning stove effect. Nevertheless, we added a few comments on the matter here as well. In particular, we better address the surface, lateral and forcing conditions of our simulations. These additions are included in more suitable Sections 3.2 and 3.3.*

**Line 153: Correct "shell" to "shall" or replace with "will"**

*It is corrected now.*

**Line 156: Is there a reference for this "own user-code"?**

*This user code is described in*

*Wolf, T., Pettersson, L. H., & Esau, I. (2020). A very high-resolution assessment and modelling of urban air quality. Atmospheric Chemistry and Physics, **20**(2), 625–647.*

*The files can be sent in response to the request.*

**Line 159: Change to "prescribed" or "prescribing"**

*It is corrected now.*

**Lines 163-164: So a bulk PM25 is emitted directly from the sources?**

*We are not sure how to understand this comment. All PM2.5 is emitted from chimneys as sources of pollution in these simulations. Each chimney adds at each time step a certain mass of PM2.5 in the grid volume where it is located.*

**Lines 164-166: How are the sources treated, as area sources per grid cell? How do you then distinguish between the "old" and "new" stoves? Is it not possible to treat the chimneys as separate sources as you have this information on 1 m resolution from DEM? Is it not possible to calculate an assign a fraction of new vs old stoves per grid cell?**

*The source treatment is described in the text with sufficient details. Each chimney adds at each time step a certain mass of PM2.5 in the grid volume where it is located. Old and new stoves are different in the amount of PM2.5 per unit time. Yes, it is possible to separate chimneys at 1 m resolution but our model is run at 10 m resolution, so we can only separate households. To avoid sharp concentration gradients – poor for numerical schemes in the model – we further aggregate the emission in 3x3 cells arrangements. It is possible to calculate the assign fraction of stoves, but we do not see any need to do this as our analysis is focused not on the situation around a specific building but at larger spatial scales.*

**Line 216: "… installation OF only the new…."**

*It is corrected now.*

**Line 216: Why do you aggregate and average in 3x3 cells rather than using the absolute values in each grid cell?**

*The use of absolute values will lead to sharp concentration gradients and numerical instability in the model.*

**Line 218: So the initial conditions are set to zero? This should be further clarified with regards to consequences. Is it not possible to use a typical "background concentration" from for example "clean days"? This also allows to more realistically evaluate the contributions of local chimneys with respect to observed levels. How about the boundary conditions? Is there transport of PM2.5 to the domain from outside?**

*The argument looks reasonable when taken into a theoretical discussion. In practice, what would a typical background concentration in a clean day? Stoves do not work and rain cleans up air. So, measurements from Måledata for luftkvalitet | NILU – Norsk institutt for luftforskning" show the concentrations of 1-2 mkg/m3 – that is insignificant for our study.*

*There is no transport of PM2.5 into the domain, and this is arguably good approximation as Bergen domain protected by mountains and open sea.*

**Line 237: Replace "correspondence" with "agreement"**

*It is corrected now.*

**Lines 235-239: It would be interesting to have lower and upper bounds of this mean and represent them in the simulations with extra scenarios.**

*We agree that such a powerful tool as PALM opens for exploration of different settings and scenarios, but such a study would be outside our present scopes.*

**Section 4.1:**
**A brief model evaluation is needed here, although earlier publications are refereed.**

*We think that additional focus on model will distract attention form the scopes of this study. We add only the following sentence:*
*The maximum simulated (observed on 11.02.2021) PM2.5 concentrations were 76.7 µg m$^{-3}$ (81.2 µg m$^{-3}$) at Danmarksplass, 53.4 µg m$^{-3}$ (59.2 µg m$^{-3}$) at Klosterhaugen in the city center, and 26.1 µg m$^{-3}$ (18.6 µg m$^{-3}$) at Rådal. This agreement demonstrates reasonably good capture of the spatial variability and accumulation of PM2.5 in the scenario simulations despite the accepted simplifications, assumptions, and uncertainties.*

**Lines 271-273: This artificial pollutant methodology is not clear and need a bit more details on what it is based on. Is it a way to transport pollutants outside the model domain or within?**

*We add more details to the description there. The methodology does not have anything to do with the transport of pollution, emission or diffusion as such. It only makes distinction between pollution from each district traceable throughout the simulations.*

**Section 4.2:**
**This section requires deeper analyses and discussion of the different mitigation scenarios. Currently, it reads like a summary of a previous study rather than stand-alone results from the present work.**

*This comment is surprising for us. We would argue that this Section presents only of the most important results of this study – a demonstration that plausible measures limited to just to some areas may eliminate the concentrations above a given threshold (Figures 6 and 7). Moreover, we present only own results between lines 330 and 355 so that about 75% of the Section total length. Nevertheless, we introduce some changes to emphasize the sovereign results of this study.*

**Conclusions:**
**The section is missing an interpretation of findings with respect to existing literature on similar works in other parts of the world in in order to put the present study in a more regional and global context. Currently, the impression is that the general interest to the results seems to be rather limited.**

*Indeed, this is important weakness to be corrected. Now, we include more on the regional and global context too, generalize the conclusions and methodology, and emphasize scientific contribution of this study to the common body of knowledge. However, there are important methodological differences that set a barrier to point-by-point intercomparison with the other studies. Previous studies did not consider specific meteorological scenarios for the highest PM concentrations. And this is for a good reason, they are based on the meteorological models that are deficient under the stably stratified atmospheric conditions, i.e., under the conditions when the highest concentrations are found. Therefore, the previous studies focus on the mean concentrations, whereas the largest (but short-term) impact is associated with the highest concentrations (e.g., Grange et al., 2013).*

*Grange, S. K., Salmond, J. A., Trompetter, W. J., Davy, P. K., & Ancelet, T. (2013). Effect of atmospheric stability on the impact of domestic wood combustion to air quality of a small urban township in winter. Atmospheric Environment, **70**, 28–38.*

---

## Author Response (AR2)

Response to referees of manuscript "Dispersion of particulate matter (PM2.5) from wood combustion for residential heating: Optimisation of mitigation actions based on large-eddy simulations" by T. Wolf, L. Pettersson and I. Esau

**Comments from the Editor**

**Thanks for your reply and the revised manuscript. Both reviewers and I are satisfied with the performed changes and your manuscript is (almost) ready for being published in ACP. There are a few more minor and technical things that should be corrected before final publication.**

We are pleased to hear this positive evaluation of our work. Indeed, we have left some technical issues to be corrected during the final submission. It is done now.

**- The referencing show frequently an inconsistent way of citing other peoples work. For example:**
**Line 100: "... as described by Maronga et al. (Maronga et al., 2015, 2019b)" should be "... as described by Maronga et al. (2015, 2019b)", same for the Wolf et al. citation that follows**
**Line 69: It should be "(Chandler, 1976; Bai, 2018)"**
**Further citation bugs can be found in lines 203, 229, 233, 358, 411, etc.**
**See guidelines here: https://www.atmospheric-chemistry-and-physics.net/submission.html#references**

We use the automated referencing in the MSWORD document. The document had some problems, and they are corrected in this submission.

**- Line 58: The word "perhaps" or the entire sentence should be removed, if you are not sure that your study is one of the first of its kind.**

We removed this ambiguity.

**- Line 83: "long-distance (in some local sense)" is a bit confusing. Do you mean "long-range or near-distant pollution transport"?**

Now, we use the term "long-range" and explain in more details what it means in the context of the local modeling study.

**- Line 142-143: "... whereas policy makers and stakeholders were less open to explore different opportunities and stove replacement strategies." This is a quite personal statement without any back-up/reference. I would suggest to remove it.**

Perhaps you are right, but such a narrow approach has been also criticized in another independent study by

Lopez-Aparicio, S., & Grythe, H. (2020). Evaluating the effectiveness of a stove exchange programme on PM2.5 emission reduction. Atmospheric Environment, 231(October 2019). doi:10.1016/j.atmosenv.2020.117529.

Nevertheless, we prefer to tone down this statement.

**- Line 183: Please put parenthesis around "(10x10x10)m^3" or write "10 m x 10 m x 10 m"**

It is corrected now.

**- Line 430 / Data availability: The need to contact the corresponding author for data is not best practice. If possible, please add the direct link or DOI to the data on the Nansen Center (see also https://www.atmospheric-chemistry-and-physics.net/policies/data_policy.html).**

We agree, a direct link to the data is now given.

**Referee #1**

Recommended acceptance.

**Referee #2**

Recommended acceptance.